# Transcriptome profiles of *Trypanosoma brucei rhodesiense* in Malawi reveal focus specific gene expression profiles associated with pathology

**Peter Nambala[1,2], Harry Noyes[3], Joyce Namulondo[4], Oscar Nyangiri[4], Vincent Pius Alibu[4], Barbara Nerima[1], Annette MacLeod[5], Enock Matovu[4], Janelisa Musaya[2], Julius Mulindwa[1]\*, on behalf of the TrypanoGEN+ Research Group as Members of the H3Africa Consortium**

1 Department of Biochemistry and Sports Sciences, College of Natural Sciences, Makerere University, Kampala, Uganda, 2 Kamuzu University of Health Sciences, Malawi-Liverpool-Wellcome Trust Clinical Research Programme, Blantyre, Malawi, 3 Centre for Genomic Research, University of Liverpool, Liverpool, United Kingdom, 4 Department of Biotechnical and Diagnostic Sciences, College of Veterinary Medicine Animal Resources and Biosecurity, Makerere University, Kampala, Uganda, 5 Wellcome Centre for Integrative Parasitology, University of Glasgow, Glasgow, United Kingdom

\* Julius.mulindwa@mak.ac.ug

**Data Availability Statement:** The phenotype and RNA sequence data have been deposited in the European Genome-Phenome Archive with

## Abstract

### Background

Sleeping sickness caused by *Trypanosoma brucei rhodesiense* is a fatal disease and endemic in Southern and Eastern Africa. There is an urgent need to develop novel diagnostic and control tools to achieve elimination of rhodesiense sleeping sickness which might be achieved through a better understanding of trypanosome gene expression and genetics using endemic isolates. Here, we describe transcriptome profiles and population structure of endemic *T. b. rhodesiense* isolates in human blood in Malawi.

### Methodology

Blood samples of r-HAT cases from Nkhotakota and Rumphi foci were collected in PaxGene tubes for RNA extraction before initiation of r-HAT treatment. 100 million reads were obtained per sample, reads were initially mapped to the human genome reference GRCh38 using HiSat2 and then the unmapped reads were mapped against *Trypanosoma brucei* reference transcriptome (TriTrypDB54_TbruceiTREU927) using HiSat2. Differential gene expression analysis was done using the DeSeq2 package in R. SNP calling from reads that were mapped to the *T. brucei* genome was done using GATK in order to identify *T.b. rhodesiense* population structure.

### Results

24 samples were collected from r-HAT cases of which 8 were from Rumphi and 16 from Nkhotakota foci. The isolates from Nkhotakota were enriched with transcripts for cell cycle

accession number EGAS00001007173, and can be accessed via the link https://ega-archive.org/search/EGAS00001007173.

**Funding:** This study was funded through the Human Heredity and Health in Africa (H3Africa; Grant ID H3A-18-004 to EM) from the Science for Africa Foundation. H3Africa is jointly supported by Wellcome Trust and the National Institutes of Health (NIH). The views expressed herein are those of the author(s) and not necessarily of the funding agencies. The funders had no role in study design, data collection and analysis, decision to publish, or preparation of the manuscript.

**Competing interests:** The authors have declared that no competing interests exist.

arrest and stumpy form markers, whereas isolates in Rumphi focus were enriched with transcripts for folate biosynthesis and antigenic variation pathways. These parasite focus-specific transcriptome profiles are consistent with the more virulent disease observed in Rumphi and a less symptomatic disease in Nkhotakota associated with the non-dividing stumpy form. Interestingly, the Malawi *T.b. rhodesiense* isolates expressed genes enriched for reduced cell proliferation compared to the Uganda *T.b. rhodesiense* isolates. PCA analysis using SNPs called from the RNAseq data showed that *T. b. rhodesiense* parasites from Nkhotakota are genetically distinct from those collected in Rumphi.

## Conclusion

Our results suggest that the differences in disease presentation in the two foci is mainly driven by genetic differences in the parasites in the two major endemic foci of Rumphi and Nkhotakota rather than differences in the environment or host response.

## Author summary

A better understanding of *Trypanosoma brucei rhodesiense* gene expression profiles and population structure using endemic isolate may speed up the search for novel diagnostic and control tools for rhodesiense sleeping sickness. Here, we analysed *T. b. rhodesiense* transcriptome profiles from endemic isolated from peripheral blood in Nkhotakota and Rumphi foci in Malawi. In Nkhotakota focus, *T. b. rhodesiense* transcripts were enriched for cell cycle arrest and stumpy marker whereas in Rumphi focus, the isolates were enriched for antigenic variation and folate biosynthesis biological pathways. Furthermore, we also found that *T. b. rhodesiense* population structure in Nkhotakota focus is different from Rumphi focus. The differences in trypanosome gene expression profiles and population structure are consistent with a less severe and acute sleeping sickness clinical profiles in Nkhotakota and Rumphi foci respectively and suggest that the differences in pathology in the two foci are mainly due to differences in the parasite rather than in the host response.

## Introduction

Human African trypanosomiasis (HAT) causes social and economic burdens on people living in remote areas where HAT is endemic. *Trypanosoma brucei gambiense* is the causative agent of gambiense HAT (g-HAT) in Western, Central and parts of Eastern Africa, whereas, *Trypanosoma brucei rhodesiense* (Tbr) causes rhodesiense HAT (r-HAT) in Southern and Eastern Africa. Although, intensified HAT surveillance championed by the World Health Organisation has led to drastic decreases in HAT incidences over the past 20 years, r-HAT is still endemic in Malawi in areas adjacent to wildlife reserves such as Nkhotakota and Rumphi districts [1]. For example, there was a sudden surge in r-HAT cases from 2019 to 2021 in Malawi's r-HAT foci and the cause is yet to be established [2]. To prevent future HAT outbreaks, there is need to develop novel epidemiological interventions and surveillance tools which might be achieved through a comprehensive understanding of the parasite genetics and gene expression profiles underlying HAT transmission cycle.

Genetic recombination between *T. b. rhodesiense* and *T. brucei brucei* during circulation in animals and tsetse vectors is believed to result in creation of new strains that may impact the epidemiological landscape of HAT diseases as it may create a genetic pool of human infective trypanosomes [3]. Genetic characterisation of Tbr isolates from Malawi and Uganda revealed that the genetic structure of trypanosomes between the two countries is different with *T. b. rhodesiense* isolates in Uganda being more clonal compared to Malawi isolates that had greater genetic diversity and evidence of frequent mating [4]. Moreover, r-HAT in Uganda tends to be an acute disease whereas it ends to be a more chronic disease in Malawi's Nkhotakota focus [5]. At the same time, genetic diversity of *T. b. rhodesiense* isolates have been observed between Uganda's three r-HAT foci with expansion of new foci from Kenya [6]. It remains unclear to what extent differences in clinical presentation are associated with parasite or host genetic diversity. We previously described human transcriptome profiles in r-HAT disease in Nkhotakota where most cases present with a chronic stage 1 disease and Rumphi where most cases present with an acute stage 2 disease [7]. We also found that there were differences in expression profiles in individuals with stage 1 and stage 2 disease but no differences in infected individuals between Nkhotakota and Rumphi foci [7].

Therefore, the current study describes transcriptome profiles and population structure of *T. b. rhodesiense* parasites isolated from r-HAT patients in Nkhotakota and Rumphi in Malawi. Additionally, we have also compared the transcriptome profiles of Tbr isolates between Malawi and Uganda. Our data shows that the differences in pathology between the two foci is associated with differences in parasite population structure. This may contribute to the search for novel trypanosome diagnostic markers and control strategies.

## Methods

### Ethics statement

Ethical approval of the study was obtained from the Malawi National Health Sciences Research Committee (Protocol Number: 19/03/2248). Written consent and assent were obtained from each study participant before sample collection.

### Study sites and sample collection

The r-HAT surveillance and study participants recruitment for blood sample collection to be used for dual RNA sequencing in this study has been previously described [2]. The samples were collected from Nkhotakota and Rumphi r-HAT foci which are approximately 400 Km apart. Briefly, sample collection was done during active and passive r-HAT surveillances conducted for 18 months from July 2019 to December 2020. HAT cases were confirmed to be infected with trypanosome parasites by microscopic examination of thick blood films during the surveillance period. 2ml whole blood samples were collected into Paxgene tubes from r-HAT cases and stored at -20°C until processing. All samples were collected before initiation of r-HAT treatment and all patients were thereafter treated following the national r-HAT treatment guidelines. A PCR targeting the serum resistance associated (SRA) gene of *T. b. rhodesiense* [8], was done on all collected whole blood samples to confirm rhodesiense HAT disease in recruited study participants.

### RNA sequencing and analysis

Dual RNA sequencing was done on the same samples that we used for human transcriptome analysis we previously described. Since trypanosomes are blood parasites it was possible to obtain trypanosome transcriptomes from the same RNA-seq data. Briefly, RNA was extracted

from blood of *T. b. rhodesiense* infected individuals using TRIzol method [9]. Samples with total RNA >1µg were selected for RNA library preparation using the QIASeq FastSelect rRNA, globin mRNA was depleted and libraries were prepared for sequencing on the Illumina NovaSeq with the NEBNext Ultra II Directional RNA Library Prep Kit to a target depth of 100 million reads. FASTQ reads were aligned to the GRCh38 release 84 human genome sequence obtained from Ensembl [10] using HiSat2 [11]. Unmapped reads were then mapped against reference transcriptome TriTrypDB54_TbruceiTREU927 using HiSat2. After mapping to the *T. brucei* transcript sequence only reads that were mapped in proper pairs were retained in the output bam-file for differential gene expression in R Studio V4.2 using DESeq2 package [12], transcripts with less than 10 reads across all samples were filtered out. All samples analyzed for RNA-seq were considered as independent biological replicates. Enriched biological pathways were determined by uploading significant (padj<0.05) differentially expressed genes in Tri-TryDB release 61 [13].

### *T.b. rhodesiense* SNP calling and analysis

The GATK workflow was used for SNP calling from reads that were mapped to the *T. brucei* transcriptome for 16 samples from Nkhotakota and 8 samples from Rumphi foci. VSG and ESAG genes were removed prior to population structure analysis since the numerous copies of these genes make mapping difficult, leading to SNP calls due to miss-alignment rather than mutation [14]. The resulting SNP data set was used in PLINK line for: i) multidimensional scaling (MDS) analysis based on raw Hamming genetic distance to generate principal coordinates as well as population cluster distance matrix and ii) for Fixation index estimation of allele variance based on Wright's F-Statistics ($F_{ST}$) [15, 16]. Graphical output of PCA and hierarchical clustering was prepared in R Studio version 4.3.1[17].

## Results

### No differences in *T.b. rhodesiense* transcriptomes from stage 1 and 2 HAT isolates

We had previously identified differences in human transcriptomes between individuals with stage 1 and stage 2 HAT disease in Malawi [7]. We used RNA-seq data from the same samples of r-HAT cases to align to the *T. brucei* transcriptome for trypanosome transcriptome analysis. A total of 16 and 8 cases were collected from Nkhotakota and Rumphi respectively. We compared transcriptomes of *T.b. rhodesiense* isolated in individuals with stage 1 and 2 HAT, no *T. b. rhodesiense* genes were significant differentially expressed (DE) between these two stages. There was also no difference in gene expression of *T.b. rhodesiense* parasites isolated from male and female HAT cases.

### Enrichment of cell cycle arrest and stumpy form marker transcripts in Nkhotakota *T.b. rhodesiense* isolates

Clinical presentation of HAT disease in Malawi is focus dependent cases presenting with stage 1 in Nkhotakota and stage 2 in Rumphi [2]. To determine the contribution of variations in trypanosome gene expression to focus specific clinical phenotypes, we performed a principal component analysis (PCA) and differential transcriptome analysis of *T.b. rhodesiense* genes of parasites from Nkhotakota and Rumphi focus. Principal components 2 and 3 separated *T.b. rhodesiense* transcriptomes from the two HAT foci into two clusters (**Fig 1A**), and a total of 91/10628 (0.86%) genes were significant (padj < 0.05) differentially expressed between the two foci. Of the 91 genes, 21/91 (23.08%) genes were upregulated log2 fold change (log2FC) >1 in

A

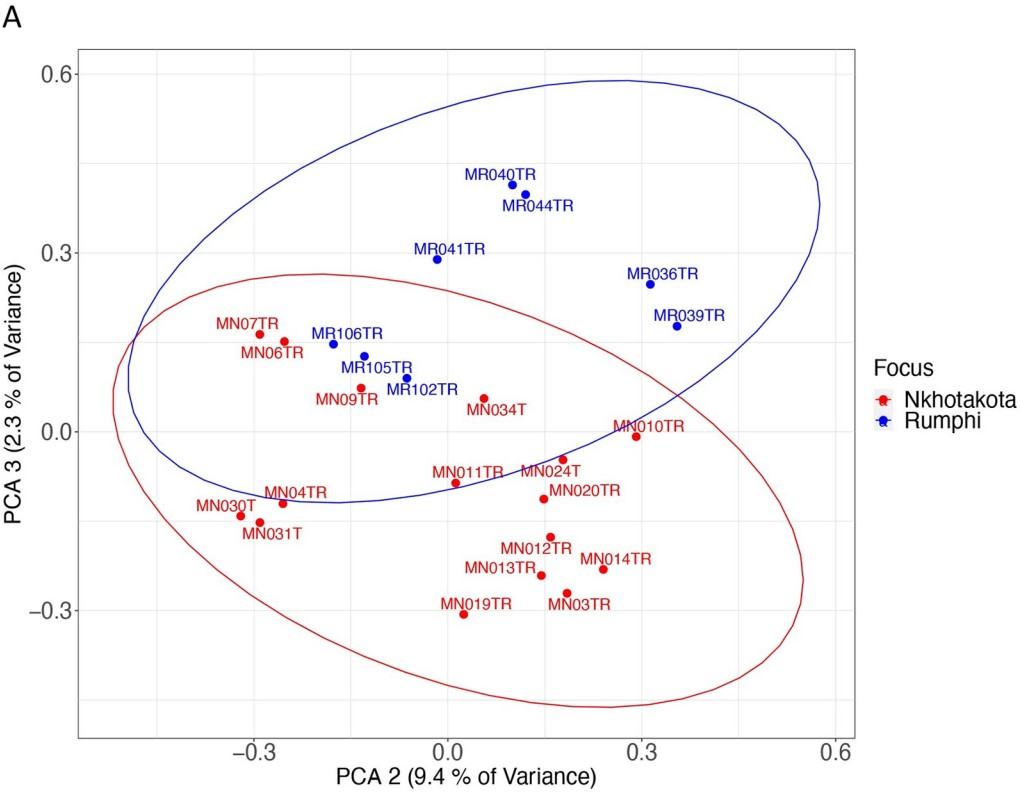

B

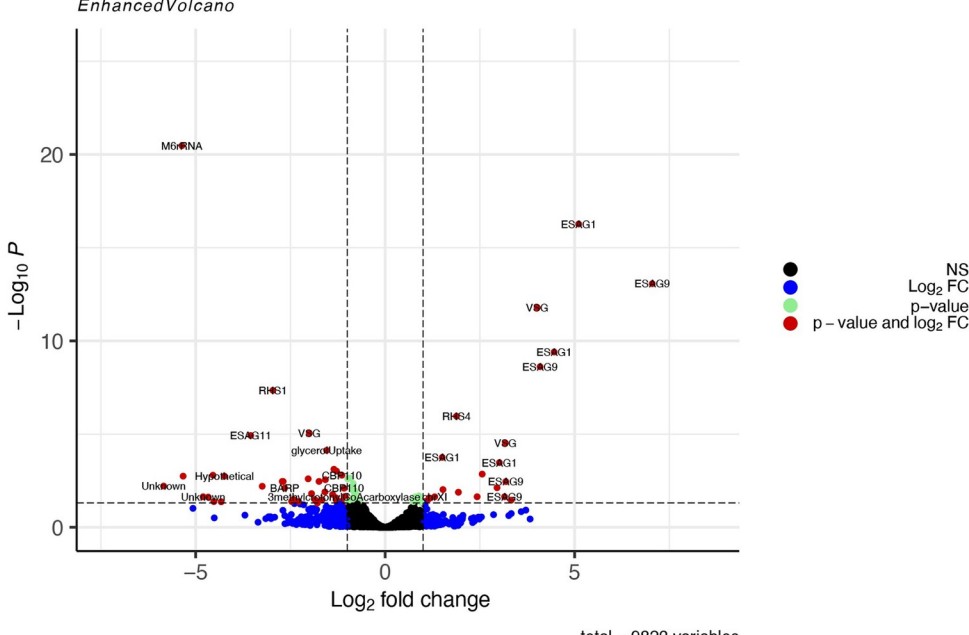

**Fig 1. A)** PCA plot showing clustering of *T.b. rhodesiense* transcriptomes in isolates from the Nkhotakota and Rumphi foci on a plot of PC2 and PC3. **B)** Volcano plot showing genes that were upregulated with log2FC >1 in Malawi *T. b. rhodesiense* isolates. Black dots represent non significant genes, blue for upregulated and downregulated genes but not significantly (padj<0.05) differentially expressed, green for highly expressed but neither upregulated nor downregulated and red for genes that were significantly differentially expressed. Genes upregulated in Nkhotakota isolates (right side) are inversely downregulated in Rumphi isolates (left side) and vice-versa.

isolates from Nkhotakota (**Fig 1B and S1 Table**), of which 43% (9/21) of the genes encode for ESAG1 (Tb09.v4.0065:mRNA, Tb927.11.17890:mRNA, Tb927.9.660:mRNA, Tb927.1.4910: mRNA), ESAG9 (Tb927.9.680:pseudogenic_transcript, Tb927.11.18670.1, Tb927.1.5220: mRNA, Tb11.1000:mRNA) and ESAG11 (Tb927.1.4900:mRNA). ESAG1 is a *T. brucei* heat shock protein that induces differentiation of procyclic trypanosomes to blood stream forms *in vitro* when there is a temperature shift from 27˚C to 37˚C [18]. ESAG 9 is a stumpy form marker that is enriched in *T. brucei* as adaptation for either tsetse vector transmission or for sustainability of infection chronicity in mammalian hosts [19, 20]. ESAG11 encodes GPI transmembrane protein which plays a role in lipid raft and glycosylation of *T. brucei* VSGs [21, 22]. Two DEGs encoded VSG (**S1B Fig**). Two invariant surface glycoproteins (Tb927.2.3310: mRNA, log2FC 1.1 and Tb927.5.620:mRNA, log2FC 0.9) were also significant differentially expressed (**S1 Table**).

To determine biological pathways exploited by *T.b. rhodesiense* during human infection in the Nkhotakota focus, upregulated genes (21/91) were loaded into TriTrypDB release 60 [13]. This showed that upregulated genes were enriched (p<9.63E-3) for endomembrane system organization, retrograde vesicle-mediated transport, Golgi organization, Golgi vesicle transport and plasma membrane organization (**Figs 1C, S2 and S2 Table**). We also observed enrichment of genes for mitotic cell cycle arrest, negative regulation of cellular amide metabolic process, negative regulation of cellular protein metabolic process and negative regulation of protein metabolic process.

Collectively, these results indicate that the *Tbr* isolates from Nkhotakota were predominantly in the stumpy form which could explain the chronic form of HAT in this focus when compared to Rumphi.

## Enrichment of folate biosynthesis and antigenic variation transcripts in Rumphi *T.b. rhodesiense* Isolates

From the 91/10628 differentially expressed genes (DEGs) in *T.b. rhodesiense* isolates, 43/91 (47.25%) genes were upregulated (log2FC > 1.0) in isolates from Rumphi focus (**Fig 2A and S1 Table**). Of the 43 genes, 14 (33%) coded for VSGs (log2FC >1.7–5.8), and 4 (9%) for ESAGs (**S4 Fig**). M6 rRNA was the most significant (padj<3.35E -25) differentially expressed gene and highly upregulated (log2FC >5) suggesting high protein synthesis in Rumphi *Tbr* isolates compared to isolates in Nkhotakota focus. Kinesin K39 was also upregulated in Rumphi parasites and is an ATP-dependent cytoskeleton motor protein which in eukaryotic cells plays a crucial role in cell cycle and migration [23]. In *Leishmania donovani*, a trypanosomatid that causes leishmaniasis, K39 kinesis accumulates and moves along the cortical cytoskeleton in a cell cycle-dependent preference for the posterior pole of the cell [24].

Next, we uploaded the 43 DEGs upregulated in Rumphi focus into TriTrypDB release 60 [13], to determine biological processes that were enriched and visualised the enriched biological processes in REVIGO [25]. For this we identified high enrichment (48 to 145 fold enrichment) of Pteridine metabolic processes and transport (**S3 Table**). Pteridine together with folic acid are essential folates used for metabolic biosynthesis of DNA, RNA and amino acids [26]. Trypanomastids exploits pteridine and folic acid metabolites in mammalian hosts and insect vectors for folate biosynthesis of purine and pyrimidine nucleotides. We also observed enrichment of *T. b. rhodesiense* biological processes involved in response and evasion of host immune response as well as in pathogen-host interaction (**Fig 2B**). These results suggest that the Tbr isolates from Rumphi were enriched for bloodstream forms that are highly replicative and exploiting the human folate metabolites for nucleotide synthesis. This could perhaps explain the observed acute nature of rHAT disease in Rhumpi as compared to Nkhotakota.

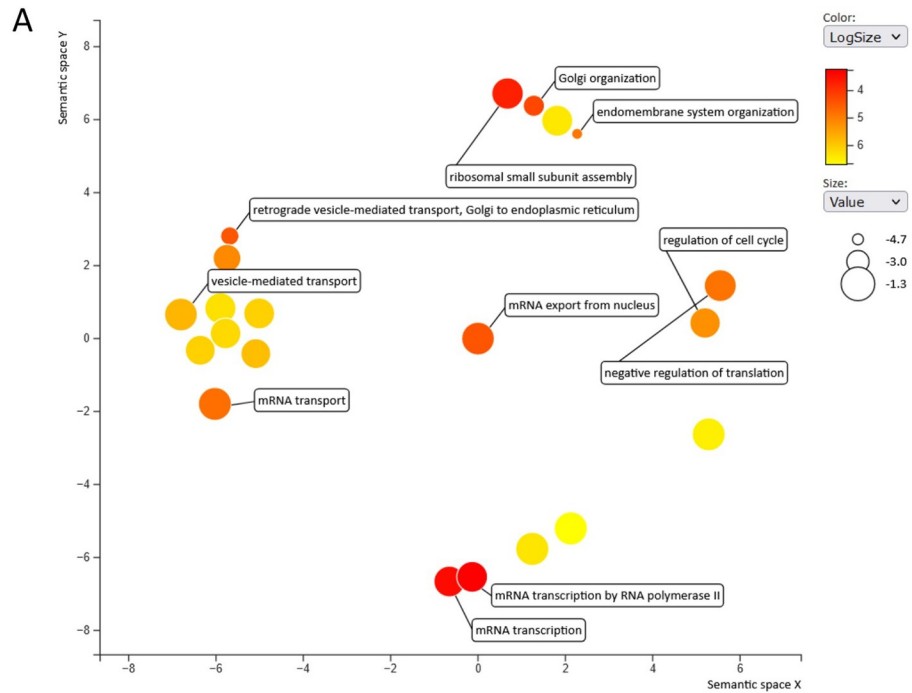

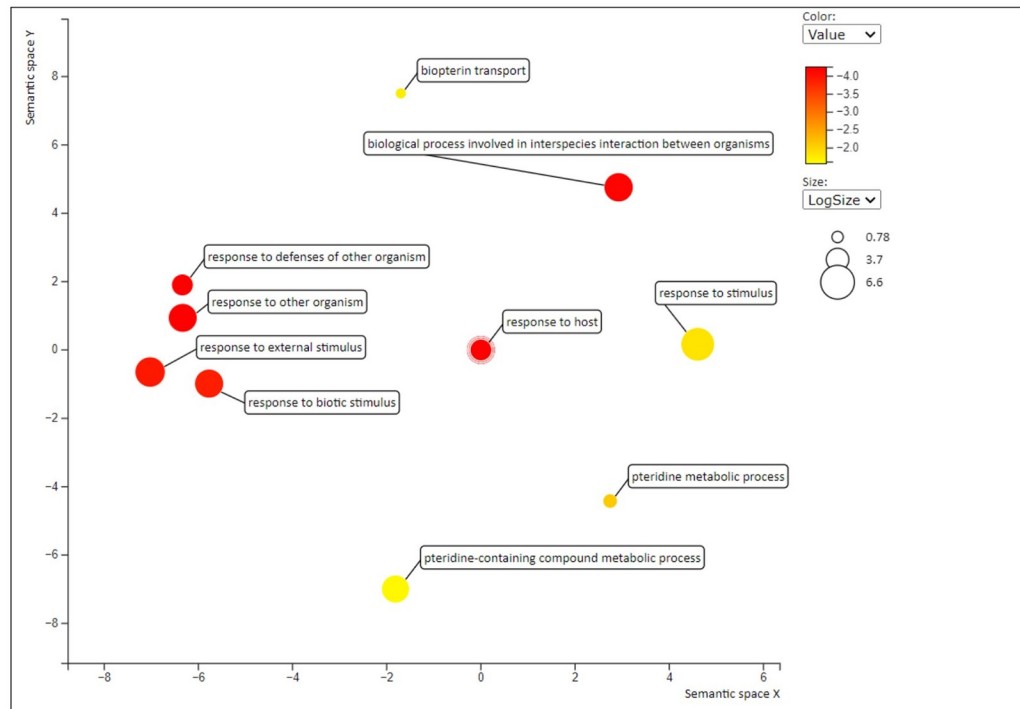

**Fig 2.** Biological Pathways of significantly differentially expressed gene **A)** Biological pathways of *T.b. rhodesiense* upregulated genes enriched during human infection in Nkhotakota focus. **B)** Biological pathways of *T.b. rhodesiense* upregulated genes enriched during human infection in Rumphi foci. Folate biosynthesis and response to host were among the enriched pathways. The axes in the plot have no intrinsic meaning but semantically similar GO terms cluster together in the plot [25].

## Malawi *T. b. rhodesiense* parasites are enriched with cell cycle arrest transcripts compared to Uganda *T. b. rhodesiense* parasites

Since *T. b. rhodesiense* gene expression is different between Malawi's r-HAT foci and clinical presentation of r-HAT varies between countries [5, 27], we next sought to compare gene expression of Malawi parasites with published data from Ugandan parasites [28, 29]. We first identified gene IDs that were mapped to both Malawi and Uganda *Tbr* isolates and filtered out gene IDs found in isolates of one country only. In total, 7003 *Tbr* Gene IDs had counts that were common to both Malawi and Uganda isolates were then loaded into DEseq2 for differential gene expression and principal component analysis. There was a distinct clustering in PCA1 and PCA3 between Malawi and Uganda human *Tbr* isolates (**S3 Fig**)**.** Parasites from Uganda also clustered together and were distinct from the Malawi parasites suggesting clonality of isolates, and isolates from rodents clustered differently from human isolates of both countries. Since rodents samples clustered differently from human samples, we excluded them from further gene expression comparison between Malawi and Uganda isolates.

A comparison of human Tbr isolates from Malawi and Uganda showed that of the 7003 transcripts detected, 3132/7003 genes were significantly expressed (padj< 0.05) in Malawi isolates, of which 2318/3132 (74.01%) were differentially expressed and 814/3132 (25.99%) genes were neither upregulated nor downregulated. Of the differentially expressed genes, 1565/2318 (67.52%) gene were upregulated (log2FC >1) and 753/2318 (32.48%) genes were downregulated (**Fig 3A and 3B**). Among the most upregulated genes were *T. brucei* Protein Associated with Differentiation (TbPAD2, log2FC 17.0) and TbPAD2 (log2FC 16.0) which are stumpy markers for blood stream trypanosomes. To identify gene ontology biological pathways enriched by the 127 genes, we uploaded the gene list in TriTrypDB release 60 and visualised in REVIGO [13]. This identified regulation of cell growth, regulation of growth, mitotic cell cycle arrest, regulation of cell population proliferation and regulation of developmental process among the enriched biological pathways for Malawi *T. b. rhodesiense* isolates compared to Uganda parasites (**Fig 3C**). This suggests that Malawi *T. b. rhodesiense* are enriched with stumpy parasites which are necessary for efficient transmission in tsetse vector compared to Uganda *T. b. rhodesiense* and could explain the more chronic nature of the Malawi strain as compared to the Uganda strain.

## Population structure and genetic diversity of *T. b. rhodesiense* isolates varies between Rumphi and Nkhotakota foci

Having identified that *T. b. rhodesiense* gene expression profiles in Malawi are focus specific, we next sought to understand whether there are differences in allele frequencies between Nkhotakota and Rumphi isolates. SNP calling from reads that were mapped to the *T. brucei* genome was done using the GATK workflow and analysed in PLINK for multidimensional scaling (MDS) analysis based on raw Hamming genetic distance [16]. The results showed a clear population stratification and genetic distance between isolates from Nkhotakota and Rumphi focus on principal components (PC) 1 and PC 2 (**Fig 4A**). The distance matrix was then used to construct a phylogenetic tree with unrooted neighbour-joining without assuming evolutionary hierarchy. The phylogenetic tree showed a clear genetic distance between *T. b. rhodesiense* populations in Nkhotakota and Rumphi focus (**Fig 4B**). To validate the genetic distance in the phylogenetic tree, we used fixation index (Fst) to measure the between groups genetic variance in *T. b. rhodesiense* populations from Nkhotakota and Rumphi. Using many SNP markers called from RNA-seq data, it is possible to get an estimate of genetic differentiation without needing to use a large sample size [30]. An Fst value of 1 suggests complete differentiation in allele frequency between subpopulations while a value of 0 suggests no

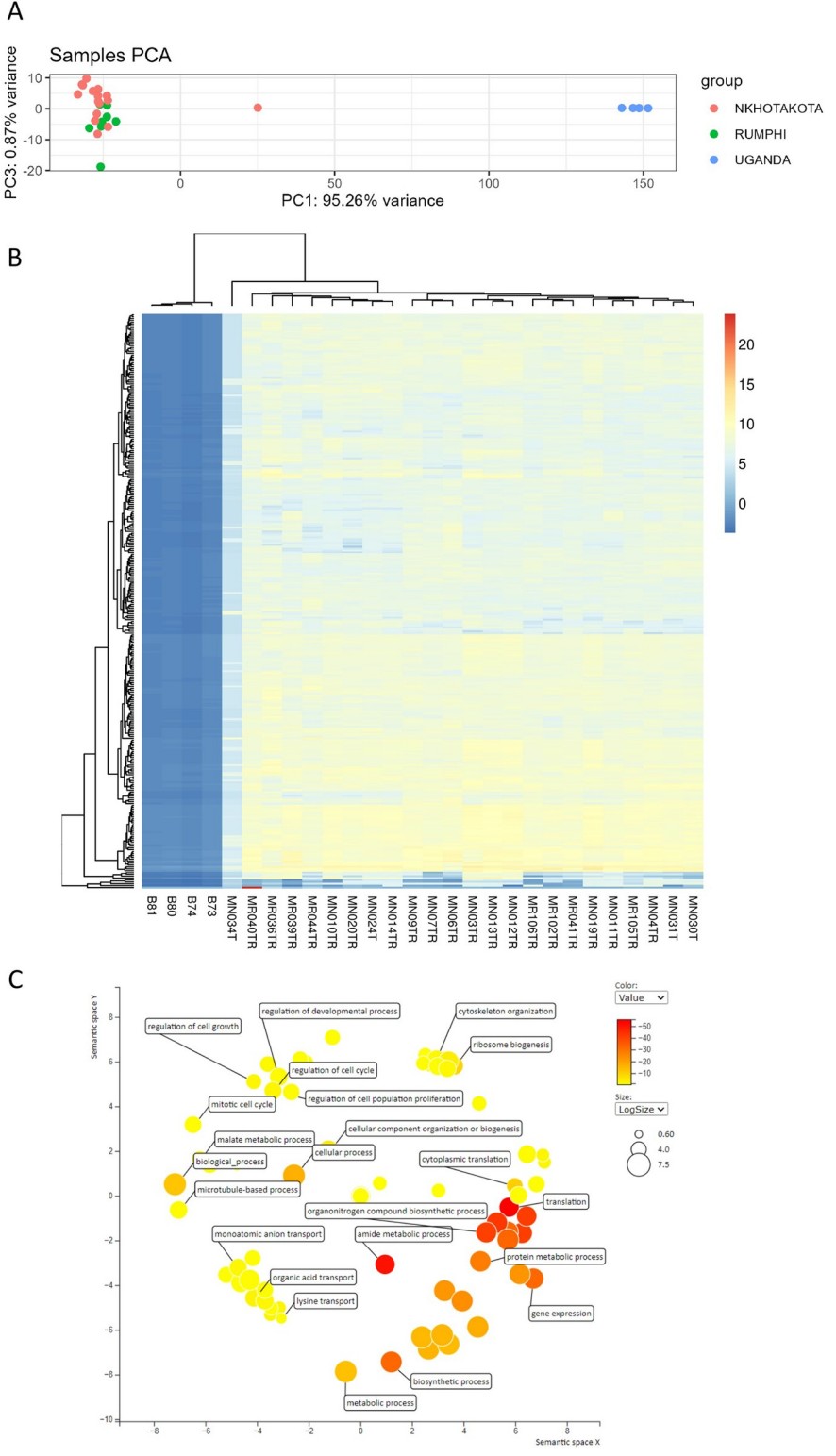

**Fig 3. Comparison of gene expression profiles between Malawi and Uganda Tbr isolates. A)** PCA analysis of Tbr isolated in individuals with stage 1 and stage 2 r-HAT in Malawi and Uganda. Sample MN034T was intermediate between Malawi and Uganda Tbr isolates **B)** A eucledian heatmap generated in PCAExplorer comparing the gene expression level of each Tbr isolate from Malawi and Uganda. Sample MN034T that was intermediate in PCA also had an intermediate gene expression levels compared to other Malawi isolates. Sample ID starting with MN, MR and B

were from Nkhotakota, Rumphi and Uganda respectively **C)** Enriched biological pathways of Malawi Tbr isolates compare to Uganda isolates loaded in TriTrypDB release 60 and visualised in REVIGO [13, 25]. Most of the transcripts were enriched for reduced cell proliferation in Malawi isolates compared to Uganda isolates.

differentiation and a mean Fst of greater than 0.15 is considered significant in differentiating populations [16]. There was a mean Fst of 0.31 between the two populations and 352 SNPs had an Fst of 1.0. The results suggest diversity in *Tbr* population structure between Nkhotakota and Rumphi focus isolates which might be due to fixation of alleles in one population or the other.

## Discussions

In the current study we have compared gene expression profiles and population structure of *T. b. rhodesiense* isolates between Nkhotakota and Rumphi foci. Additionally, we have compared gene expression profiles between Malawi and Uganda, representing Southern Africa and Eastern Africa respectively. Upregulation of VSGs in Rumphi isolates suggest high proliferation of bloodstream slender form trypanosomes compared to isolates in Nkhotakota. Slender Trypanosomes use a repertoire of VSGs to evade host adaptive immune system whereas, stumpy trypanosomes have much lower levels of VSG expression [14, 20, 31]. This is consistent with differential expression of kinesin K39 which maintains cell cytoskeleton integrity during the cell cycle. Moreover, high human antibody titre against *Leishmania chagas* and *Leishmania donovani* kinesin K39 antigen has been detected in patients with Chagas disease making kinesin K39 a potential biomarker for serological diagnosis [32]. Identification of kinesin K39 in *T. b. rhodesiense* isolates highlights this as a potential biomarker for a much-needed serodiagnosis of *T. b. rhodesiense* infections and should be validated in future studies.

We have also identified high enrichment of transcripts for pteridine in *T. b. rhodesiense* isolates form Rumphi focus. Structural differences between the fusion protein DHFR-ThyS in trypanosomatids and the individual polypeptides in humans make this protein (folate) an attractive target for rational drug design which should be exploited in future research. Additionally, exploitation of host folate metabolites by *T. b. rhodesiense* isolate may have implications of clinical pathology of r-HAT as it may induce host anaemia when the parasite uses haemoglobin as a folate source. Consistent with this finding, studies on clinical presentation of HAT in Malawi had identified anaemia as one of the main clinical pathologies associated with HAT patients [33].

The contrasting enrichment of genes required for adaption of *T. b. rhodesiense* transmission to tsetse fly vector and sustainability of blood stream slender trypanosomes in Nkhotakota and Rumphi foci may have implications on r-HAT clinical phenotype, control and elimination. Indeed, we previously established that most r-HAT cases in Nkhotakota present with a stage 1 disease whereas in Rumphi most r-HAT cases present with a severe stage 2 r-HAT disease [2]. Virulence of trypanosome infection in mammalian host is determined by accumulation of a population of slender trypanosomes [34]. We speculate that severe acute cases of r-HAT observed in Rumphi in comparison to Nkhotakota, might be due to the ability of isolates in Rumphi to maintain high population of slender trypanosomes whereas those in Nkhotakota foci are predominantly the non-dividing stumpy forms and hence lower parasitaemia. We further propose that *T. b. rhodesiense* isolates in Nkhotakota focus might be highly transmissible as they increase expression of stumpy markers whereas isolates in Rumphi focus maybe less transmissible due to high maintenance of a slender trypanosome population during human infections. Nonetheless, future research should consider validating our current findings using appropriate experimental models.

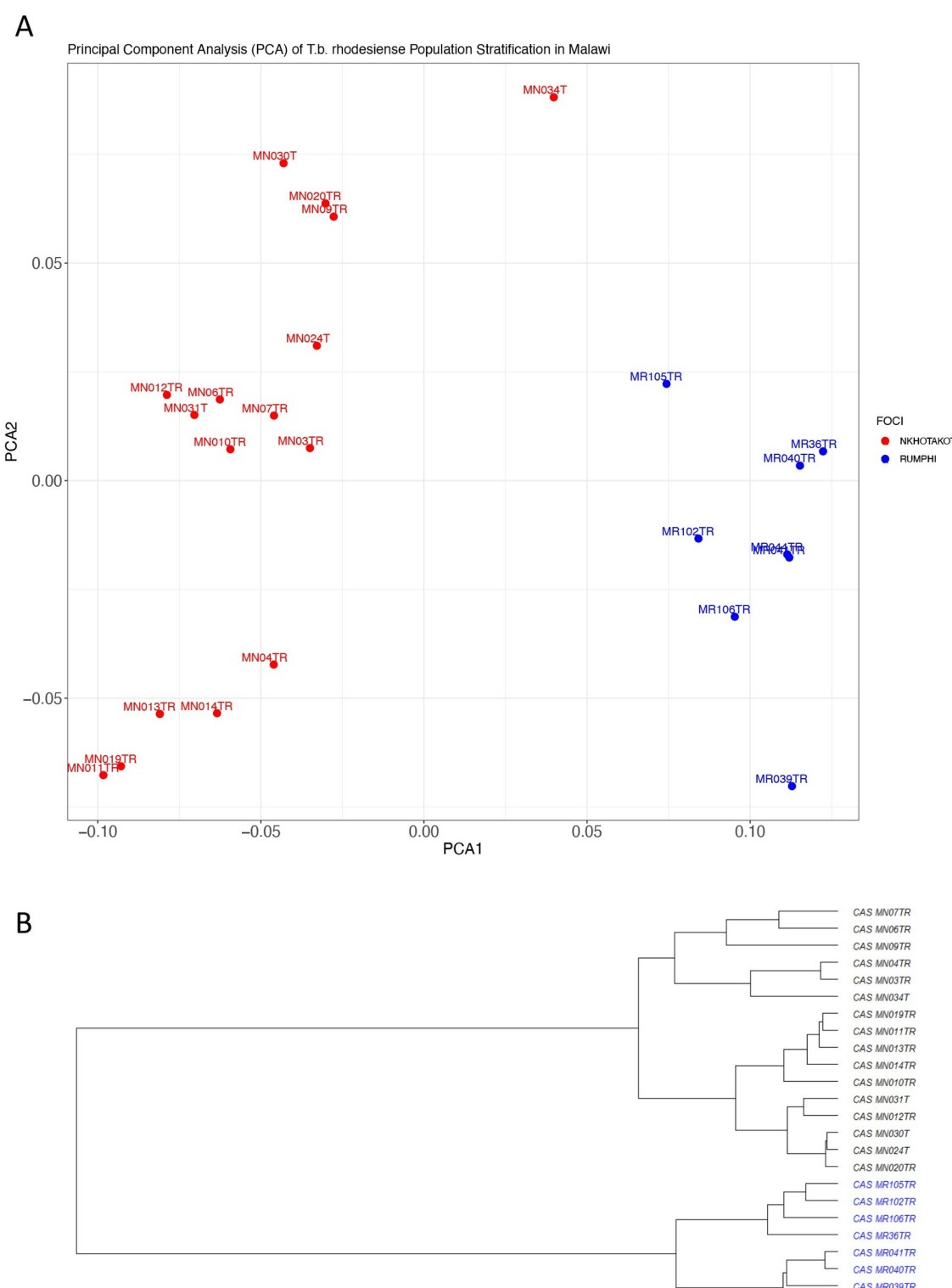

**Fig 4. Polulation structure and genetic diversity of *T. b. rhodesiense* between Nkhotakota and Rumphi foci based on SNPs extracted from RNA-seq data. A)** population clustering of isolates from Nkhotakota and Rumphi foci on Principal Component Analysis (PCA) 1 and PCA 2. Red and aqua colors represents isolates from Nkhotakota and Rumphi foci respectively. **B)** Unscaled hierarchical clustering dendogram showing the relatedness of Tbr isolates from Nkhotakota and Rumphi foci that was generated using population cluster distance matrix. Blue and black color represents isolates from Rumphi and Nkhotakota respectively.

Comparison of gene expression profiles between Malawi (Southern Africa) and Uganda (East Africa) showed distinct clustering of samples between the two countries which is consistent with microsatellite analysis results of isolates from Malawi and Uganda [4, 6]. Additionally, previous population genetics analysis identified that Ugandan isolates have a clonal population compared to diversified Malawi (Nkhotakota focus) isolates which was consistent with our transcriptomics results. Enrichment of cell cycle arrest biological pathways in Malawi isolates demonstrates the need for control strategies to focus on breaking the contact cycle between humans and tsetse fly vectors in Malawi r-HAT foci. Stratification of samples from human and rodents also suggests that *T. brucei* exploits different genes when circulating in dissimilar mammalian hosts. Inference from animal models or cell culture results on disease in humans should be done with caution as it may not be a true representation of *T. b. rhodesiense* infection dynamic in humans.

We have also identified 2 and 14 VSGs in Nkhotakota and Rumphi isolates respectively, that were significant differentially expressed and upregulated in *T. b. rhodesiense* in all blood samples analysed. Blood samples used in the current study were randomly collected over a period of 18 months excluding the possibility that the expressed VSG were randomly expressed. Although most expressed VSGs are highly antigenic and constantly changing, some VSG also elicit little host antibody response thereby subverting natural immunity [35]. Additionally, two invariant glycoproteins were also significant differentially expressed in isolates from Rumphi and Nkhotakota foci. The identified invariant glycoproteins and VSGs, as cell surface proteins, have a potential to be explored in future research to determine if they continue to be consistently expressed, if they are they could be used as biomarkers for a much-needed rapid diagnostic test and vaccine against *T. b. rhodesiense* infections. In animal trypanosomes, a cell surface protein localized to the flagella has been used to develop a vaccine candidate that offered protection against *T. vivax* infection in mice models [35].

In conclusion, our results from both gene expression profiles and population genetic analysis have added new insights on how clinical phenotypes of r-HAT might be influenced by differences in *T. b. rhodesiense* population structure and gene expression profiles. We have used RNA-seq data to call *T. b. rhodesiense* SNPs from endemic isolates which will contribute to future studies of *T. b. rhodesiense* population genetic studies. SNP analysis results showed a distinct stratification in *T. b. rhodesiense* population structure between isolates from Nkhotakota and Rumphi foci suggesting that there is little mixing of parasites between these two foci and that therefore there is potential to control the infection in each focus independently. This is consistent with the results we obtained from trypanosome gene expression profiles which showed distinct clustering of gene expression profiles of isolates from each focus. Additionally, we have showed that transcriptome profiles of *T. b. rhodesiense* isolates in Nkhotakota and Rumphi are different. Peripheral blood parasites in Nkhotakota, where most cases are detected at stage 1, were enriched with transcripts for stumpy trypanosomes, whereas in Rumphi, where most cases are detected in stage 2, the transcripts were enriched for antigenic variation and folate biosynthesis. Lastly, we have also identified differences in transcriptome profiles between Malawi and Uganda *T. b. rhodesiense* isolates. Transcriptomes of Malawi *T. b. rhodesiense* were enriched for cell cycle arrest compared to Uganda isolates. Future research should consider validating our findings by obtaining pathological markers in rodents infected with *T. b. rhodesiense* isolates from Nkhotakota and Rumphi foci.

In this study, we analysed trypanosome transcriptomes in blood without comparing with cerebral spinal fluid (CSF) which would have provided more insight on the parasite's gene expression regulation when circulating in CSF as well. Due to resource limitations, the study used single replicates for RNA sequencing instead of triplicates for control of possible technical

variations. Furthermore, it would not be ethical to collect samples at multiple time points prior to treatment, so we will remain dependent on animal studies for understanding the transcriptome over time. Additionally, future studies should also consider using a larger sample size for population genetics of *T. b. rhodesiense* isolate in Nkhotakota and Rumphi foci as we only managed to sequence 24 samples. A more in-depth analysis of the differentially expressed ESAGs should also be considered to assess whether they are likely to be functionally redundant or not of which was beyond the objective of this study. Nonetheless, our findings have brought new insight in *T.b. rhodesiense* gene expression and population genetics in endemic samples which may contribute to the future development of novel control strategies, diagnostic tools and vaccine candidates.

## Supporting information

**S1 Table. List of Genes that were significant (padj<0.05) differentially expressed in T.** brucei rhodesiense isolates from Nkhotakota focus versus Rumphi focus.
(DOCX)

**S2 Table. Significant (p<0.05) gene ontology enrichment of T.b.** rhodesiense biological processes of differentially enriched genes (DEGs) that were upregulated (log2FC > 1) in Nkhotakota focus and loaded in TritrypDB. The fold enrichment is the percentage of genes loaded divide by the percentage of genes with this term in the background. The p-value measured the Fishers exact test.
(DOCX)

**S3 Table. Significant (p<0.05) gene ontology (GO) enrichment of T.b.** rhodesiense biological processes of differentially enriched genes (DEGs) that were upregulated (log2FC > 1) in Rumphi focus and loaded in TritrypDB. The fold enrichment is the percentage of genes loaded divide by the percentage of genes with this term in the background. The p-value measured the Fishers exact test.
(DOCX)

**S1 Fig. Differential gene expression analysis of Tbr isolates from Nkhotakota versus Rumphi foci. A)** PCA analysis showing outlier samples from both Nkhotakota and Rumphi foci that were excluded from further analysis. **B)** Proportions of upregulated differentially expressed genes in in Nkhotakota Tbr isolates with ESAG transcripts being the most upregulated.
(TIFF)

**S2 Fig. Fold enrichment of gene ontology biological processes of Nkhotakota Tbr isolates loaded in TriTrypDB [13].**
(TIFF)

**S3 Fig. Principal component analysis of Tbr transcriptomes comparing human isolates from Nkhotakota (16 isolates), Rumphi (Eight isolates), Uganda (Four isolates) and Tbr isolates passaged in rodents.**
(TIFF)

**S4 Fig. Significant differentially expressed VSGs in Malawi *T. b. rhodesiense* isolates. A to B)** Normalised counts of VSGs differentially expressed in isolates from Nkhotakota foci. **C to P)** Normalised counts of VSGs differentially expressed in isolates from Rumphi foci.
(JPG)

## Acknowledgments

We would like to acknowledge Nkhotakota and Rumphi district health offices for their assistance in sample collection.

## Author Contributions

**Conceptualization:** Peter Nambala, Harry Noyes, Annette MacLeod, Enock Matovu, Janelisa Musaya, Julius Mulindwa.

**Formal analysis:** Peter Nambala, Harry Noyes, Joyce Namulondo, Oscar Nyangiri, Julius Mulindwa.

**Funding acquisition:** Annette MacLeod, Enock Matovu.

**Investigation:** Peter Nambala, Julius Mulindwa.

**Methodology:** Peter Nambala, Harry Noyes, Julius Mulindwa.

**Supervision:** Enock Matovu, Janelisa Musaya, Julius Mulindwa.

**Validation:** Julius Mulindwa.

**Writing – original draft:** Peter Nambala.

**Writing – review & editing:** Harry Noyes, Vincent Pius Alibu, Barbara Nerima, Annette MacLeod, Enock Matovu, Janelisa Musaya, Julius Mulindwa.

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
