## [Decision Letter · Decision Letter 0]

21 Aug 2023

Dear Dr Mulindwa,

Thank you very much for submitting your manuscript "Transcriptome profiles of T.b. rhodesiense in Malawi reveal focus specific gene expression Profiles associated with pathology." for consideration at PLOS Neglected Tropical Diseases. As with all papers reviewed by the journal, your manuscript was reviewed by members of the editorial board and by several independent reviewers. In light of the reviews (below this email), we would like to invite the resubmission of a significantly-revised version that takes into account the reviewers' comments. 

This is an interesting study that makes a contribution to explaining the differences in clinical presentation of natural infections of sleeping sickness in different foci in Malawi. The authors need to pay close attention to the comments by reviewers, including:

Considering the small number of samples, the exclusion of 4 samples is not well-justified. The reader would like to see where there fall, for example in the PCA analysis. 

The quality of figures is poor and should be improved.

All other comments.

We cannot make any decision about publication until we have seen the revised manuscript and your response to the reviewers' comments. Your revised manuscript is also likely to be sent to reviewers for further evaluation.

Sincerely,

Daniel K. Masiga

Academic Editor

Abhay Satoskar

Section Editor

This is an interesting study that makes a contribution to explaining the differences in clinical presentation of natural infections of sleeping sickness in different foci in Malawi. The authors need to pay close attention to the comments by reviewers, including:

Considering the small number of samples, the exclusion of 4 samples is not well-justified. The reader would like to see where there fall, for example in the PCA analysis. 

The quality of figures is poor and should be improved.

All other comments.

Reviewer's Responses to Questions

**Key Review Criteria Required for Acceptance?**

**Methods**

-Are the objectives of the study clearly articulated with a clear testable hypothesis stated?

-Is the study design appropriate to address the stated objectives?

-Is the population clearly described and appropriate for the hypothesis being tested?

-Is the sample size sufficient to ensure adequate power to address the hypothesis being tested?

-Were correct statistical analysis used to support conclusions?

-Are there concerns about ethical or regulatory requirements being met?

Reviewer #1: The objectives are clear and the hypothesis testable. Since the study essentially examines two spatially seprated populations (Nkhotakota and Rumphi foci), more details on descriptions of these populations and a spartial map where the samples were collected and probably barriers between these 'populations' would help. Secondly, it not clear from the writing on how many replicates were used in RNA Seq... where 16 and 8 cases were collected from Nkhotakota and Rumphi respectively considered as independent replicates? This needs to be clarified. For the population genetics component, the numbers in each population is not clear. Typically a sample size of 30 and above is considered sufficient. More details is required here.

Reviewer #2: (No Response)

Reviewer #3: Yes.

**Results**

-Does the analysis presented match the analysis plan?

-Are the results clearly and completely presented?

-Are the figures (Tables, Images) of sufficient quality for clarity?

Reviewer #1: Appropriate data collection and analyses approaches were employed and ethical approval obtained. The quality of the figures need improvement. Line 186 -"..genes were upregulated (padj> 1.0)...." padj should be <0.05 for significance. The data has not also been deposited in EGA or any other public repository yet, which in turn limits efforts of validating the findings.

Reviewer #2: The analysis presented match the analysis plan and the results are clearly presented except that the figures does not have sufficient quality for clarity.

Reviewer #3: Line 159-160: “Four samples were identified as outliers in the PCA and were excluded from the analysis (Fig S1A).” 

Can the authors clearly mark the outliers in Fig S1A and justify why they are considered outliers and not the real representation of the population diversity?

Lines 165 – 171: On ESAGs being DE: one must be careful inferring from these results because ESAGs belong to multicopy gene families often characterized by redundant genes. So, whilst the upregulation of particular non-redundant ESAG copies may be of relevance, upregulation of redundant ESAGs may just be the product of chance. My suggestions to address this issue are either to acknowledge this limitation in text, or to perform a more in-depth analysis of the DE ESAGs (by phylogenetics for example) to assess whether they are likely to be functionally redundant or not. 

Line 171 and 187: 

VSGs being DE may just mean that the two populations were expressing different superabundant VSGs at the time of collection. Were these VSGs equally abundant across all samples of the same group? Or is one sample contributing more than others? 

Fig 1B and Fig 2A are the same, just flipped. I suggest merging Fig 1 and 2 together, and identifying in Fig 1B which side of the graph represents upregulation in Rumphi and which side represents upregulation in Nkhotakota. It would also help to plot Log2 FC against average expression, side by side with Fig. 1B.

Line 219: If rodent samples were not included in the analysis, why add them in the first place? It confuses the reader.

Fig 3B: What are the labels of each sample? I recommend adding information on the origin of each sample as in the PCA plot. 

Lines 221-224: The number of DE genes don’t seem to add up. The authors mention a total of 3132 DE genes, but only 1565 upregulated and 753 downregulated in Malawi, leaving 814 genes missing.

Lines 226-228: are those genes highly upregulated also highy abundant? Small changes in genes expressed at very low levels result in very high fold changes, not necessarily relevant.

Fig 4C is not present in my version of the manuscript and therefore I could not review it (referred in lines 255-257).

All figure legends need further detail.

Line 264: VSGs were not overexpressed, but rather upregulated. This needs to be confirmed as it would only be relevant if their expression level (ultimately represented by read counts rather than fold change) is high enough. 

Line 267: stumpy forms do express VSGs. However, due to their cell arrest state, they have overall much lower leves of mRNA. 

Line 309-311: one would not expect to find the same VSGs expressed in different samples, especially not in samples from distinct strains, hence the beauty and complexity of antigenic variation. A good example for the situation described in this paragraph would have to be based on an invariant gene and not on a variant antigen. 

Lines 314-317: This is a very interesting finding in the paper. Please elaborate on the expression level of those VSGs and their nature. The authors state they’re unique: does this mean they were never seen in published genomes? What is their degree of nucleotide identity with closest relatives in Tb927? Were they present in the Ugandan samples? Can the authors present the sequences as a supplementary file? 

Out of curiosity, how far is Rumphi from Nkhotakota? I’d add this information to the manuscript. It helps to have an idea if a lot of crossover is expected or not.

**Conclusions**

-Are the conclusions supported by the data presented?

-Are the limitations of analysis clearly described?

-Do the authors discuss how these data can be helpful to advance our understanding of the topic under study?

-Is public health relevance addressed?

Reviewer #1: the conclusions are essentially supported by the data presented. However, since the findings have not been validated (are putative) some of the conclusions should be toned down to reflect this observation.

Reviewer #2: There are a few places in the discussion that should be revised as indicated in the attachment

Reviewer #3: Yes.

**Editorial and Data Presentation Modifications?**

Reviewer #1: minor revision.

Reviewer #2: (No Response)

Reviewer #3: line 83: Tbr isolates, define

line 87: typo, tends to be 

Lines 224 and 226 appear as strikethrough in my version. 

Line 469: typo, axes

Lines 192-194 are better suited for discussion. 

Whenever figures are taken from tritrypdb, please add that information to fig legends (e.g. Fig S1C).

Line 491: typo, dendrogram

Fig 4B: explain which phylogenetic method was used in fig. legend.

I suggest switching paragraph “Malawi T. b. rhodesiense parasites are enriched with cell cycle arrest transcripts compared to Uganda T. b. rhodesiense parasites” with paragraph “Population Structure and Genetic diversity of T. b. rhodesiense isolates Varies Between

Rumphi and Nkhotakota Foci”, as to continue the comparison within Malawi before moving to a comparison across countries.

Line 269: typo, Leishmania chagasi. Also, L. infantum should be the name used in my (and others) opinion based on the law of priority. Please see: https://www.scielo.br/j/mioc/a/z7Y47Dk5Hs33SCbFvc53JXF/?lang=en

Line 309: dynamics (plural) seems more appropriate here

**Summary and General Comments**

Reviewer #1: Most of the novel findings came from the studies related to comparison of the two populations in Malawi, the comparison between malawi and Uganda appears focused on validating previous findings. Save for the reservations I have highlighted above, the experiments were well designed and executed, with the findings providing insights on further studies that can be performed to validate the outcomes.

Reviewer #2: (No Response)

Reviewer #3: The authors present a competent analysis of T. brucei rhodesiense transcriptomes from two disease foci in Malawi to investigate whether observed differences in clinical presentation could be explained by parasite genetic diversity and/or gene expression diversity. They also compare those data with previously published T. brucei rhodesiense transcriptomes from Uganda. 

They found that transcriptomes from Malawi isolates are largely distinct from Uganda, but they also identify considerable differences in isolates from different Malawi foci. 

They conclude they parasite diversity might play a role in clinical presentation, a hypothesis that requires experimental validation.

The results from the DE analysis suggests that parasites from foci where chronic disease is more common express genes involved in transmission to higher levels, whereas parasites from foci where acute disease is prevalent express genes more associated to antigenic variation and folate metabolism. Whilst I agree with the major findings and methods of the study, I recommend a few alterations to the text and data presentation, as well as to conduct a more detailed analysis particularly in terms of VSG expression.

PLOS authors have the option to publish the peer review history of their article (what does this mean?). If published, this will include your full peer review and any attached files.

Reviewer #1: No

Reviewer #2: No

Reviewer #3: No
---

## [Decision Letter · Decision Letter 1]

10 Jan 2024

Dear Dr Mulindwa,

Thank you very much for submitting your manuscript "Transcriptome profiles of T.b. rhodesiense in Malawi reveal focus specific gene expression Profiles associated with pathology." for consideration at PLOS Neglected Tropical Diseases. As with all papers reviewed by the journal, your manuscript was reviewed by members of the editorial board and by several independent reviewers. The reviewers appreciated the attention to an important topic. Based on the reviews, we are likely to accept this manuscript for publication, providing that you modify the manuscript according to the review recommendations. 

Sincerely,

Daniel K. Masiga

Academic Editor

Abhay Satoskar

Section Editor

Reviewer's Responses to Questions

**Key Review Criteria Required for Acceptance?**

**Methods**

-Are the objectives of the study clearly articulated with a clear testable hypothesis stated?

-Is the study design appropriate to address the stated objectives?

-Is the population clearly described and appropriate for the hypothesis being tested?

-Is the sample size sufficient to ensure adequate power to address the hypothesis being tested?

-Were correct statistical analysis used to support conclusions?

-Are there concerns about ethical or regulatory requirements being met?

Reviewer #1: The methods and well described and are reproducible.

Reviewer #2: Authors have sufficiently addressed the concerns in the revised manuscript. I do not have any additional comments.

Reviewer #3: (No Response)

**Results**

-Does the analysis presented match the analysis plan?

-Are the results clearly and completely presented?

-Are the figures (Tables, Images) of sufficient quality for clarity?

Reviewer #1: The results are well presented and tally with the methods presented earlier

Reviewer #2: Authors have sufficiently addressed the concerns in the revised manuscript. However the quality of the figures should be improved as it's difficult to read most of the legends.

Reviewer #3: (No Response)

**Conclusions**

-Are the conclusions supported by the data presented?

-Are the limitations of analysis clearly described?

-Do the authors discuss how these data can be helpful to advance our understanding of the topic under study?

-Is public health relevance addressed?

Reviewer #1: Conclusions are well supported by data.

Reviewer #2: Authors have sufficiently addressed the concerns in the revised manuscript. I do not have any additional comments.

Reviewer #3: (No Response)

**Editorial and Data Presentation Modifications?**

Reviewer #1: accept

Reviewer #2: Authors have sufficiently addressed the concerns in the revised manuscript. I do not have any additional comments.

Reviewer #3: (No Response)

**Summary and General Comments**

Reviewer #1: while my concerns have been addressed. I am still concerned about the use of single replicate in the analysis..

Reviewer #2: Authors have sufficiently addressed the concerns in the revised manuscript. I do not have any additional comments.

Reviewer #3: Line 239-244: “A comparison of human Tbr isolates from Malawi and Uganda showed that 3132/7003 (44.72%) gene were significantly (padj< 0.05) differentially expressed of which 1565/3132 (49.97%) gene were upregulated in Malawi (log2FC >1), 753/3132 (24.04%) genes were downregulated, and 814 genes were neither upregulated nor downregulated (Fig 3A and 3B).”

I appreciate the effort made by the authors, but this is still not correct. 

If there are 3132 differentially-expressed genes, they must all be either up- or down-regulated. I think the authors are assuming differential expression based on just the adjusted p-value. This is not standard procedure, we should use both LFC and adj p-value as thresholds to consider a gene differentially expressed. Based on the values presented, the total number of DE genes should be 1565+753=2318. The authors should also clearly state the number of transcripts detected in total, which I suppose were 7003, and update the percentages since the total should no longer be 3132. 

Line 289: “Slender Trypanosomes use a repertoire of VSGs to evade host adaptive immune system whereas, stumpy trypanosomes have much lower levels of VSG expression”

A reference should be added here. 

Lines 322-326: “We further propose that T. b. rhodesiense isolates in Nkhotakota focus might be highly transmissible as they overexpressed stumpy markers whereas isolates in Rumphi focus maybe less transmissible due to high maintenance of a slender trypanosome population during human infections.”

As addressed in a previous comment, overexpressed is not the correct term here. Overexpression relates to forced expression in a mutant line. The correct term is upregulation or simply “they increase expression of”.

Fig 1B 

- title: T. b. rhodesiense requires spaces and to be italicized.

- It’s not clear the directionality of the DE analysis. In other words, which side represents upregulation in Rumphi and which side is in Nkhotakota?

- Typo in legend: vice-versa

PLOS authors have the option to publish the peer review history of their article (what does this mean?). If published, this will include your full peer review and any attached files.

Reviewer #1: No

Reviewer #2: No

Reviewer #3: No

Figure Files:

Data Requirements:

Reproducibility:

References

---

## [Editor Report · Decision Letter 2]

26 Feb 2024

Dear Dr Mulindwa,

Thank you very much for submitting your manuscript "Transcriptome profiles of Trypanosoma brucei rhodesiense in Malawi reveal focus specific gene expression profiles associated with pathology." for consideration at PLOS Neglected Tropical Diseases. As with all papers reviewed by the journal, your manuscript was reviewed by members of the editorial board and by several independent reviewers. The reviewers appreciated the attention to an important topic. Based on the reviews, we are likely to accept this manuscript for publication, providing that you modify the manuscript according to the review recommendations. 

Sincerely,

Abhay R Satoskar

Section Editor

Abhay Satoskar

Section Editor

Figure Files:

Data Requirements:

Reproducibility:

References

---

## [Editor Report · Decision Letter 3]

17 Apr 2024

Dear Dr Mulindwa,

We are pleased to inform you that your manuscript 'Transcriptome profiles of Trypanosoma brucei rhodesiense in Malawi reveal focus specific gene expression profiles associated with pathology.' has been provisionally accepted for publication in PLOS Neglected Tropical Diseases.

Best regards,

Shaden Kamhawi

Editor in Chief

---

## [Editor Report · Acceptance letter]

29 Apr 2024

Dear Dr Mulindwa,

We are delighted to inform you that your manuscript, "Transcriptome profiles of <i>Trypanosoma brucei rhodesiense<i> in Malawi reveal focus specific gene expression profiles associated with pathology.," has been formally accepted for publication in PLOS Neglected Tropical Diseases.

Best regards,

Shaden Kamhawi

co-Editor-in-Chief

Paul Brindley

co-Editor-in-Chief
